# HIV-Associated Dermatological Alterations: Barrier Dysfunction, Immune Impairment, and Microbiome Changes

**DOI:** 10.3390/ijms26073199

**Published:** 2025-03-30

**Authors:** Muhammad Anshory, Handono Kalim, Jan L. Nouwen, Hok Bing Thio

**Affiliations:** 1Department of Dermatology, Erasmus MC, University Medical Centre, 3015 GD Rotterdam, The Netherlands; m.anshory@erasmusmc.nl; 2Department of Internal Medicine, Faculty of Medicine, Universitas Brawijaya, Malang 65145, Indonesia; hkalim333@gmail.com; 3Department of Medical Microbiology and Infectious Diseases, Erasmus MC, University Medical Centre, 3013 GD Rotterdam, The Netherlands; j.l.nouwen@erasmusmc.nl

**Keywords:** HIV, skin changes, immune system, skin microbiome, stress and environment

## Abstract

Human Immunodeficiency Virus (HIV) significantly impacts skin structure, immune responses, and the microbiome, contributing to diverse dermatological conditions. The epidermis, a key physical and immunological barrier, undergoes structural changes such as hyperplasia and inflammatory infiltrates. Skin adnexal structures like hair follicles also play a role in immune modulation but are affected by HIV-related disruptions. Innate and adaptive immune systems are compromised due to CD4+ T-cell depletion, cytokine imbalances, and altered immune regulation, leading to conditions such as hypersensitivity and inflammatory dermatoses. The skin microbiome in HIV patients shows distinct shifts, including reduced Cutibacterium species and increased opportunistic microbes, independent of CD4+ levels. Age, sex, and environmental stressors exacerbate these changes, with women exhibiting stronger immune responses but higher risks of autoimmune diseases and aging men experiencing accelerated immunosenescence. Understanding these interconnected alterations is essential for developing targeted therapies to manage skin complications and improve the overall health of HIV patients.

## 1. Introduction

The skin is a vital organ that plays a dual role in protecting our body, acting as both a physical barrier and a regulator of immune responses. On the outside, it shields us from harmful agents like microbes, allergens, and environmental hazards, which can result in infections or trigger inflammatory reactions. The outermost layer of the skin, the epidermis, serves as the first line of defense, preventing these external threats from penetrating deeper into the body. Internally, the skin is integral to immune system regulation. By simultaneously preventing external harm and modulating internal immune activity, the skin is indispensable to overall health and immune defense mechanisms [1].

Human Immunodeficiency Virus (HIV), a systemic infection, profoundly affects the immune system and the skin. In its early stages, HIV often manifests through specific skin conditions such as herpes zoster, seborrheic dermatitis, and papular pruritic eruptions. These skin symptoms are among the first sign of infection and inflammation, and reveals the virus’s impact on immune function. HIV compromises the immune system, increasing vulnerability to opportunistic infections and inflammation. Even individuals without a prior history of type 2 inflammation may develop exaggerated immune responses due to the immune dysregulation caused by the virus. This heightened susceptibility and the presence of skin conditions reflect the complex interplay between HIV and the immune system. Understanding these interactions is critical for early diagnosis, effective management, and improving the quality of life for people living with HIV [2].

Stigmatization is another significant challenge for individuals with HIV, particularly when visible skin conditions are present. HIV is often perceived more negatively than other illnesses due to its strong association with “promiscuous” behavior, resulting in societal stigma and fear. Anticipation of stigma can deter individuals from disclosing their HIV status, fearing rejection and social isolation. While direct experiences of stigma may be less frequent, those who disclose often face distancing from loved ones and, at times, discriminatory treatment from healthcare providers. Internalized stigma exacerbates psychological distress, leading to feelings of despair, self-devaluation, and even suicidal thoughts. These challenges, combined with the visible impact of HIV-related skin conditions, significantly complicate the lives of those affected. Addressing these issues requires a holistic approach that integrates medical care, psychological support, and societal efforts to reduce stigmatization, ultimately enhancing the well-being of individuals living with HIV [3].

## 2. Skin’s Components and Its Change in HIV Patients

### 2.1. The Skin and Its Defence Mechanism

The skin acts as a vital defense mechanism, protecting the body from external threats through its complex, multi-layered structure and specialized components. The epidermis serves as the body’s primary barrier, protecting against physical, chemical, and infectious agents. Besides, it contains antimicrobial molecules and Langerhans cells for antigen sampling, along with CD103+ T cells and *γδ* T cells for targeted immune responses, ensuring robust defense and skin health [4]. Cutaneous appendages like hair, sebaceous glands, and sweat glands maintain skin homeostasis. Hair follicles, formed from epidermal invagination, regulate immune responses and stem cell activity. Sebaceous glands secrete lipids that support barrier function and influence the microbiome, while sweat glands aid in thermoregulation and hydration. Dysregulation of these glands is linked to skin disorders like acne and psoriasis [5] Antimicrobial peptides (AMPs) and skin lipids form a defense system against microbes. AMPs, produced by keratinocytes and other cells, combat pathogens and modulate immunity. Skin lipids, stored in the stratum corneum and secreted by sebaceous glands, are metabolized into antimicrobial molecules, enhancing protection. Together, AMPs and lipids maintain skin health and immunity [4].

A study by Chawhan et al. on HIV patients found myxoid dermal changes unrelated to specific lesions. Herpes zoster showed nuclear changes in epidermal and adnexal structures, with lymphocytic infiltrates and epidermal hyperplasia common across lesions. These findings highlight the skin’s complex response to infections [6]. There are also changes and hypertrophy in the dermis and epidermis, leading to xeroderma or xerosis. This occurs due to an impairment in the barrier function of the stratum corneum, which results from a decrease in lipid content within the viable epidermis and dermis, as well as an excessive concentration of carotenoids, particularly lycopene, within the epidermis. This dermatologic condition is a primary cause of premature skin aging in HIV patients and one of the contributing factors to pruritus in these individuals [7].

### 2.2. Mucosal Structure and Its Susceptibility to HIV

The mucosal epithelium varies by anatomical location, forming a critical barrier against HIV transmission. In the oral mucosa, keratinization levels differ by region; masticatory areas like the gingiva and hard palate have a thick keratin layer, reducing antigen entry, while mobile or specialized structures, such as lingual papillae, feature non-keratinized or mixed epithelia. Saliva plays a critical role in oral defense, containing several antiviral components with specific anti-HIV properties, including cathelicidins, defensins, lactoferrin, lysozyme, and secretory leukocyte protease inhibitor (SLPI) [8]. The oral mucosa’s thickness and the absence of HIV-susceptible immune cells in superficial layers further lower transmission efficiency. Tight and adherent junctions in the stratified squamous epithelium enhance this barrier, preventing viral paracellular entry [9,10].

HIV infection often alters the composition and functionality of saliva, reducing levels of key components like lactoferrin and Immunoglobulin A (IgA) [8]. This alteration is frequently linked to *Candida* infections in HIV patients. Lactoferrin and lysozyme, which play a vital role in innate immunity, typically increase during the early stages of HIV infection but decline as the disease progresses. However, even elevated levels of lactoferrin may not prevent candidiasis, likely due to the emergence of *Candida* species that have developed resistance to lactoferrin and lysozyme as HIV advances [11]. On the other hand, IgA, particularly secretory IgA (S-IgA), has been proposed as a novel approach for the diagnosis and therapy of HIV patients [12]. S-IgA, which is predominantly found in the gastrointestinal system, works alongside innate immunity to combat oral infections. Its unique characteristics also make it a promising candidate as a vaccine carrier for HIV antigens [13,14].

Similar to oral mucosa, the anogenital mucosa exhibits structural variations that significantly influence HIV susceptibility. In the female genital tract, the lower tract (vagina and ectocervix) features a stratified epithelium that sheds to prevent pathogen colonization, while the upper tract (endocervix, endometrium, and fallopian tubes) consists of a single-layer columnar epithelium with tight junctions. The transformation zone, where these epithelia meet, is immunologically active but highly vulnerable to HIV due to the presence of abundant immune cells. In males, the foreskin’s outer keratinized layer provides a protective barrier, whereas the inner thin, weakly keratinized layer is more susceptible to viral invasion. Similarly, the anorectal mucosa, composed of stratified squamous anal epithelium and single-layer rectal epithelium, is prone to micro-abrasions during intercourse, particularly at the transformation zone, making it a critical site for HIV entry. Structural changes during sexual intercourse further heighten susceptibility, especially in the vaginal wall and ectocervix [9].

Unlike gastrointestinal secretions, where S-IgA predominates, human semen, cervicovaginal secretions, and urine contain higher levels of IgG than IgA. In female genital secretions, immunoglobulin levels are strongly influenced by hormonal fluctuations, varying significantly throughout the menstrual cycle. Approximately half of these immunoglobulins are produced locally by plasma cells in the genital mucosa, with the remainder originating from the bloodstream. This contrasts sharply with the intestinal tract, where over 90% of immunoglobulins are produced by plasma cells in the gut’s lamina propria. As a result, systemic immunization with HIV-1 antigens can induce HIV-1-specific antibodies in genital secretions but fails to protect the intestinal tract, leaving it vulnerable. These findings underscore the need for targeted immunization strategies to elicit humoral immune responses that effectively block HIV-1 entry at both genital and intestinal mucosal sites [10].

### 2.3. Skin Microbiota and the Change of Population

The human microbiome, enriched by advances in sequencing technologies like 16S rRNA gene analysis, highlights the body as a complex multi-species ecosystem. Skin microbiota, primarily comprising *Firmicutes*, *Bacteroidetes*, *Proteobacteria*, and *Actinobacteria*, play a vital role in skin health. While strain-level identification remains challenging, differences between strains can significantly impact immunity. Other microorganisms, including viruses, fungi, and parasites, also contribute to the skin’s immune functions. Microbiome composition varies by environment—sebaceous, moist, or dry—and is influenced by health, behavior, and external factors. For example, *Staphylococcus epidermidis* enhances barrier function and protects against pathogens through antibacterial peptides and immunomodulation. Together, the microbiota and innate immune system form an essential network for skin immunity and homeostasis [15]. A recent study by Gribonika et al. found that the skin functions as an autonomous lymphoid organ, maintaining host–microbiota symbiosis. Exposure to new skin commensals triggers two parallel responses regulated by Langerhans cells: classical germinal centers in lymph nodes producing IgG1/IgG3, and tertiary lymphoid organs in the skin sustaining IgG2b/IgG2c [16]. Regulatory T cells convert into T follicular helper cells to support this process. Autonomous skin antibody production controls local microbial biomass and prevents systemic infections, revealing compartmentalized humoral responses that balance symbiosis and pathogen defense [16].

In people living with HIV, the skin microbiome undergoes notable changes, including an increase in microbes such as *Micrococcus* and *Kocuria* and a marked reduction in *Cutibacterium* species [17]. Although, this population can be varied in different diseases and skin conditions [18,19]. These shifts occur independently of CD4+ T cell levels, suggesting mechanisms beyond immune cell depletion may be driving these alterations. While the precise causes remain unclear, these changes may play a role in the development or exacerbation of skin-related conditions associated with HIV. Understanding these dynamics could provide valuable insights for developing targeted and safe treatments for microbe-associated skin disorders, highlighting the importance of the skin microbiome in overall immune function and disease management [17].

### 2.4. Innate and Adaptive Immune System of the Skin

The innate immune system of the skin forms the first line of defense against pathogens, relying on an array of phagocytic cells, innate leukocytes, and keratinocytes (Figure 1). Phagocytes like macrophages, neutrophils, and dendritic cells, alongside natural killer (NK) cells, mast cells, basophils, and eosinophils, collaborate to detect and respond to threats. Epidermal keratinocytes also play an active role, releasing inflammatory cytokines and antimicrobial molecules upon sensing pathogen-associated molecular patterns (PAMPs) or danger-associated molecular patterns (DAMPs). This detection is mediated by pattern-recognition receptors (PRRs) such as toll-like receptors (TLRs), nucleotide-binding oligomerization domain-like receptors (NLRs), retinoic acid-inducible gene I-like receptors (RLRs), and c-type lectin receptors (CLRs). These PRRs recognize specific pathogens—such as *TLR2*, *TLR6*, and *NOD2* (nucleotide-binding oligomerization domain-2) for *Staphylococcus aureus*—and trigger inflammatory cascades, recruiting neutrophils for immediate pathogen clearance. As the response progresses, macrophages shift from a pro-inflammatory (M1) to an anti-inflammatory (M2) phenotype, facilitating tissue repair and wound closure [20].

The strength and specificity of the innate immune response significantly influences the subsequent adaptive immune response. Dendritic antigen-presenting cells (APCs) in the epidermis (Langerhans cells) and dermis (dermal dendritic cells) bridge innate and adaptive immunity by presenting antigens to T and B lymphocytes. B cells recognize soluble protein antigens directly, while T cells require antigens to be presented as peptides on MHC molecules. CD8+ T cells interact with endogenous antigens via major histocompatibility complex (MHC) I molecules, while CD4+ T cells are activated by exogenous peptides presented on MHC II molecules. Major APCs include dendritic cells, B cells, and macrophages, which process and present these peptides to facilitate precise immune responses [21].

HIV disrupts this intricate immune system at multiple levels, targeting key immune cells such as Langerhans cells and dermal dendritic cells at mucosal sites. T cell helper (Th) 17 cells, vital for maintaining epithelial barrier integrity, are depleted in the gastrointestinal tract early in HIV infection, though evidence of their depletion in the skin remains limited. HIV exploits dendritic cells for viral transmission, while macrophages act as reservoirs for latent infection. Keratinocytes, though not directly infected, secrete cytokines that enhance viral replication. Mast cells, expressing *CCR5* and *CXCR4*, can harbor latent HIV, contributing to the complexity of the infection. Additionally, untreated HIV patients often exhibit exaggerated non-IgE-mediated responses to certain drugs, such as ciprofloxacin, through mast cell activation by Mas-related G-protein coupled receptor member X2 (MRGPRX2)—a receptor that can also be triggered by the HIV-1 TAT protein fragment [22].

A hallmark of HIV infection is the depletion of CD4+ T cells, leading to reduced proliferative capacity, increased expression of inhibitory molecules like cytotoxic T-lymphocyte-associated antigen 4 (CTLA-4) and programmed death 1 (PD-1), and heightened apoptosis rates (Figure 1). This depletion causes a shift from Th1 to Th2 cytokine polarization, reducing cytotoxic T lymphocyte activity and increasing IL-4, IL-5, and IgE levels [23]. Conversely, HIV infection triggers an expansion of CD8+ T cells, which, while attempting to control the virus, can induce tissue damage and exacerbate skin conditions. For instance, CD8+ T-cell-mediated granulysin secretion can result in keratinocyte death. Tissue-resident memory (TRM) cells, known to mediate antiviral responses in other infections, might also play a role in HIV-related skin disorders, though evidence remains sparse. These cells may cross-react with drug-induced peptides, leading to hypersensitivity reactions. The role of immunosuppressive regulatory T cells (Tregs) in HIV-related skin lesions is similarly unclear; while Tregs are often decreased in conditions like psoriasis and toxic epidermal necrolysis (TEN), their frequency and function in HIV remain inconsistent across studies. T cell exhaustion also played a role in the HIV immunologic shift, as several studies already showed that CD4+, CD8+ and interferon-responsive CD8+ T cells’ exhaustion occurs in HIV patients [24]. This was also showed by an increase in *PD-1* and *Tim3* as markers of T cell exhaustion [25,26].

Together, these disruptions highlight the multifaceted ways HIV undermines skin immunity, contributing to disease progression and associated complications [22].

## 3. Emerging Diseases Related to HIV Infection

### 3.1. Inflammation and Infection

HIV patients often experience a range of inflammatory diseases, some of which can serve as early indicators of HIV infection. These conditions include, but are not limited to, atopic dermatitis, psoriasis, lichen planus, seborrheic dermatitis, pruritic papular eruption, and even simple pruritus [27]. These inflammatory issues may either worsen pre-existing skin conditions or emerge after the individual contracts HIV, as is often the case with psoriasis [28]. Recognizing the relationship between HIV and these dermatological conditions is essential for early detection and effective management of the disease [27].

The pathophysiology behind these inflammatory reactions in HIV patients is complex and multifactorial. Some diseases may occur as initial sign of HIV, while others can present after a decline of CD4+ count or following antiretroviral treatment (ART) as immune reconstitution inflammatory syndrome [27]. One key theory suggests that a significant reduction in CD4+ T cells triggers a shift toward a Th2 immune response, which increases the risk of developing atopic disorders. This immune shift can compromise the skin’s barrier function, leading to a reduction in epidermal lipid content and dry skin [29], even in individuals with no prior history of atopic conditions. In addition, this immune imbalance may amplify inflammatory responses, contributing to the wide array of dermatological issues commonly seen in those living with HIV. Understanding these mechanisms is critical for developing targeted treatments to manage these skin complications effectively [30].

Another contributing factor to these inflammatory responses is the imbalance between CD4+ and CD8+ T cells during HIV infection. Since CD4+ T cells are the primary targets of the virus, their numbers decrease significantly, resulting in an inverted CD4:CD8 ratio. This imbalance disrupts normal immune system function, which may facilitate the development and exacerbation of inflammatory and atopic conditions. The reduced number of CD4+ T cells weakens the immune response, making the body more susceptible to infections and inflammatory diseases. Recognizing the impact of this immune imbalance is vital for creating strategies aimed at restoring immune function and effectively managing the associated inflammatory skin conditions in HIV patients [31].

### 3.2. Hypersensitivity

As previously mentioned, in advanced stages of HIV infection, a shift towards a Th2 cytokine profile is observed, which leads to increased infiltration of monocytes, eosinophils, and B-cell activation. This shift results in heightened immune activation and dysfunction of immunoregulatory mechanisms, contributing to the development of inflammatory and hypersensitivity skin disorders [22]. The underlying pathophysiology of these skin conditions is thought to be multifactorial, involving a complex interplay of metabolic, immunologic, host, and viral factors. Persistent damage to HIV-infected cells, immune dysregulation, increased oxidative stress, depletion of immunoregulatory cells, and consumption of protective antioxidant molecules are believed to generate excessive ‘danger signals.’ These signals trigger a cascade of immune responses, cytokine release, and hypersensitivity reactions, further exacerbating skin inflammation [32].

As HIV infection progresses, the loss of CD4+ T cells, expansion of CD8+ T cells, and chronic immune activation become prominent features. This chronic immune activation is associated with increased production of pro-inflammatory cytokines, such as *IP-10*, *MIG*, *TNF-α*, IL-6, *IFN-α*, and IL-10. Recent studies suggest that genetic polymorphisms in cytokine genes may influence the susceptibility of HIV patients to drug-related hypersensitivity reactions, including those triggered by efavirenz [32].

In cases of drug-related hypersensitivity in HIV patients, multiple mechanisms come into play. The exact pathway through which drugs provoke immune responses remains unclear, with two main hypotheses: the hapten-dependent and hapten-independent pathways. The hapten-dependent hypothesis suggests that most drugs are chemically inert until metabolized into reactive intermediates, which can bind covalently to proteins, forming haptenated complexes that are then presented to T cells via human leucocyte antigen (HLA) molecules. In contrast, the hapten-independent or pharmacological interaction hypothesis proposes that the parent drug itself directly interacts with T cells through an MHC-restricted pathway, independent of metabolism. Some studies support this hypothesis, showing that T cells from allergic individuals proliferate in vitro when exposed to the drug. Additionally, the ‘danger hypothesis’ suggests that hypersensitivity reactions may require co-stimulatory signals, such as cytokines, in addition to antigen presentation [33].

During the acute phase of drug hypersensitivity, such as with co-trimoxazole, T cells infiltrate the skin and secrete cytokines like IL-5, granzyme, and eotaxin, which promote the recruitment and differentiation of eosinophils. CD4+ T cells also play a role in hypersensitivity reactions to drugs like carbamazepine, with IL-8, a chemokine that attracts neutrophils and induces cell death, being involved in conditions like acute generalized exanthematous pustulosis. T cells can also mediate target cell killing through the perforin pathway. CD8+ T lymphocytes are particularly responsible for bullous reactions such as Stevens–Johnson syndrome (SJS) and toxic epidermal necrolysis (TEN), but they are also implicated in abacavir hypersensitivity. A crucial factor in the severity of drug hypersensitivity reactions is individual susceptibility, particularly the role of specific HLA alleles in the response to HIV medications [33].

## 4. Sex, Age, and Environmental Effects

### 4.1. Men and Women Differences

Females exhibit stronger innate and adaptive immune responses than males, resulting in more effective infection clearance and lower infection-related mortality. These differences are driven by variations in immune cell composition—females have higher numbers of CD4+ T cells, B cells, and Tregs, while males have more natural killer (NK) cells with reduced effector function—and are further shaped by sex hormones. Estrogens regulate immune genes, enhance or suppress inflammatory responses depending on context, and promote stronger type I interferon responses, while androgens generally suppress inflammation by modulating TLR expression, cytokine production, and central tolerance mechanisms like AIRE and FOXP3 expression in the thymus [34]. These hormonal and genetic factors not only influence general immune function but also play a critical role in sex-specific responses to infections like HIV-1.

HIV-1 disproportionately affects women, particularly in sub-Saharan Africa, where they account for 60% of cases, with young women aged 15–24 being significantly impacted. Gender inequalities, including limited access to education and healthcare, gender-based violence, and economic disparities, exacerbate vulnerabilities. Biologically, women exhibit distinct immune responses, such as lower viral loads and higher CD4+ T-cell counts compared to men, yet they face faster progression to AIDS at similar viremia levels, potentially due to stronger immune activation driven by sex hormones like estrogen and genetic factors such as X-chromosomal gene expression. Sex hormones also modulate HIV-1 pathogenesis, with estrogen enhancing immune defenses but also increasing immune activation and inflammation, contributing to chronic comorbidities like cardiovascular disease. TLR activation and interferon-stimulated gene expression further influence outcomes in women, highlighting the need for targeted research on hormonal and genetic contributions to HIV pathogenesis [35].

Sex-based differences significantly influence HIV acquisition, progression, and treatment. Women’s higher susceptibility to male-to-female transmission stems from biological and immune factors, while immune system distinctions, such as lower baseline inflammation and unique responses to ART and pre-exposure prophylaxis (PrEP), affect treatment efficacy. Hormonal contraceptives, particularly depot medroxyprogesterone acetate (DMPA), alter immune and microbiome environments, impacting HIV risk. Additionally, genetic and epigenetic factors, including X-chromosomal gene expression, shape immune responses and disease outcomes. Despite these complexities, cure research often overlooks sex differences, although latent reservoir studies suggest distinct dynamics in women. Addressing these gaps through inclusive, sex-specific research is critical to optimizing prevention, treatment, and cure strategies for both sexes [36,37].

Regarding disease-specific prevalence, gender-based differences were noted in the distribution of skin manifestations. Females exhibited a significantly higher occurrence of esophageal candidiasis, bacterial infections, herpes simplex, dermatophytosis, dry skin and seborrheic dermatitis, whereas males showed a notably higher prevalence of Kaposi’s sarcoma, scabies, psoriasis, and leishmaniasis [29,38,39]. Numerous studies indicate that dermatological side effects and adverse reactions to antiretroviral therapy (ART) can vary significantly between men and women. These differences were particularly pronounced with first- and second-generation antiretrovirals, as women were more likely to experience skin rashes and lipodystrophy. These adverse effects contributed to higher rates of treatment discontinuation and nonadherence among women compared to men [35,40,41].

### 4.2. Age Differences

Age-related changes in the immune system, known as “immunosenescence”, involve a dysregulation of both innate and adaptive immune functions, making older adults more vulnerable to infections, autoimmune diseases, and cancer. This process leads to a reduced ability to mount effective immune responses to infections and vaccinations. A key feature of immunosenescence is “inflammaging”, a chronic, low-grade inflammation marked by an increase in pro-inflammatory cytokines (IL-1, IL-6, TNF-α) and a decrease in anti-inflammatory cytokines (IL-10, TGF-β). Inflammaging occurs in the absence of infection and contributes to biological aging and the development of age-related diseases. As immune cells age, they adopt a “senescence-associated secretory phenotype” (SASP), releasing factors that perpetuate inflammation, increasing the risk of diseases and premature mortality. These immune changes are further exacerbated by lifestyle factors such as smoking, obesity, alcohol use, physical inactivity, and UV exposure [42].

However, recent perspectives on immunosenescence and inflammaging have shifted, recognizing that these processes may also be linked to longevity. Age-related thymic involution, which limits the T-cell receptor repertoire, might reduce energy consumption, thus supporting other essential body functions. Some evolutionary theories suggest that immunosenescence serves as an optimization strategy, balancing resources in the aging body despite its potential to contribute to disease. An alternative view suggests that immunosenescence is marked by a decline in adaptive immunity, while inflammaging reflects the activation of innate immunity, creating a feedback loop that sustains chronic inflammation. Additionally, a newer hypothesis proposes that these immune changes may be the body’s adaptive response to chronic stress, with outcomes that can either promote healthy longevity or lead to pathological aging and associated diseases, depending on genetic and environmental factors [42].

This interplay between immunosenescence and chronic stress becomes particularly significant when considering the challenges posed by HIV and aging. In HIV patients, inflammaging can occur even in younger individuals, as premature aging is hypothesized in this group due to persistent immune activation and chronic inflammation [43]. This phenomenon is thought to arise from several mechanisms linked to HIV infection, including DNA damage, telomere attrition, loss of proteostasis, mitochondrial dysfunction, cellular senescence, and stem cell exhaustion. These systemic changes can have widespread effects on patients, potentially impacting various aspects of their health, including their skin condition [44]. Thus, the convergence of immunosenescence, chronic stress, and HIV-related mechanisms underscores the complex interplay between aging and disease progression in this population.

In sub-Saharan Africa, where populations face the dual burden of both chronic diseases and infectious diseases like HIV/AIDS, the aging process in HIV-infected individuals introduces complexities that exacerbate health risks. A study by Parikh, et al. revealed significant clinical and immunological differences in older adults starting ART, with older individuals exhibiting higher systolic blood pressure, lower creatinine clearance, and higher CD4+ cell counts compared to their younger counterparts [45]. While the higher CD4+ counts and viral loads in older adults suggest earlier access to HIV care, they also point to the complexities of aging with HIV, including the potential for immune system decline and poorer clinical outcomes in this population. These findings underscore the importance of developing integrated healthcare strategies for older adults that address both HIV and aging, with a particular focus on gender issues and the unique health needs of this group [45].

### 4.3. Sex and Age Combination Effect

Sex differences in immunity persist throughout life, becoming more pronounced with aging and influencing both immunosenescence and health outcomes. Older men tend to exhibit higher monocyte activity, greater expression of myeloid cell-related genes, and a faster decline in B and T cell functions, while women experience slower immunosenescence, which correlates with longer life expectancy but often poorer health in later years. Aging female T cells produce more IL-10, mitigating inflammaging, and women demonstrate stronger humoral responses, which contribute to their higher prevalence of autoimmune diseases, with 80% of such conditions occurring in women at earlier ages. Studies show that aging primarily impacts CD8+ cells, while sex differences specifically affect CD4+ cells, macrophages, and immune gene expression, with gene activation and deactivation patterns differing between sexes. Notably, women respond better to immunostimulating therapies, emphasizing the need for personalized health and vaccination strategies that consider sex-based immune variations [42].

### 4.4. Effect of Environmental and Psychological Stress

Climate change-related and human-driven environmental changes, such as air pollution, extreme heat, UV exposure, wildfires, storms, and cigarette smoke, have significant consequences for the immune system of the skin [46]. The skin acts as a protective barrier against environmental damage, but exposure to air pollutants triggers oxidative stress and inflammation, leading to both skin and systemic pathologies. These effects are exacerbated by chronic UV radiation exposure [47]. Additionally, these environmental factors disrupt the skin microbiota, further compromising the skin barrier [48]. Such disturbances can cause or worsen conditions like premature aging, skin cancer (e.g., melanoma, SCC, BCC), inflammatory diseases (e.g., atopic dermatitis, psoriasis), acne, alopecia, pigmentary disorders (e.g., vitiligo, melasma), and itching disorders [47].

In addition to environmental stressors, psychological stress also plays a critical role in immune dysregulation. Acute stress temporarily enhances immune cell activity and increases pro-inflammatory cytokines, but chronic stress leads to prolonged inflammation, which raises the risk of chronic diseases such as atherosclerosis and frailty. It also accelerates immune system aging, making the body more vulnerable to illnesses. In individuals with autoimmune diseases, chronic stress can worsen symptoms by amplifying immune activation and hindering immune regulation. This dysregulation of the immune system, fueled by both environmental and psychological stress, contributes to poorer health outcomes, particularly in vulnerable populations [49].

For HIV patients, chronic stress worsens the immune response, accelerating HIV progression and increasing the likelihood of neuropsychiatric disorders such as depression, anxiety, and post-traumatic stress. Stress and HIV together contribute to cognitive impairment and systemic inflammation, making HIV patients more susceptible to infections and comorbidities, even with viral suppression through ART. HIV-related neuropsychiatric disorders are more common in HIV patients than in the general population, impacting ART adherence and overall quality of life. These stress-induced conditions, compounded by factors such as financial strain, food insecurity, and early life stress, underscore the need for comprehensive care that addresses both the psychological and physical aspects of stress [50]. A study by Leserman et al. found that faster progression to AIDS was linked to several factors, including a higher cumulative average of stressful life events, coping through denial, elevated serum cortisol levels, and lower cumulative satisfaction with social support. In contrast, other factors such as age, education, tobacco use, and risky sexual behavior did not significantly predict disease progression. The risk of developing AIDS approximately doubled with a 1.5-unit decrease in average support satisfaction, a single severe stressor, a one-unit increase in denial, or a 5 mg/dL rise in cortisol levels [51]. This study is supported by more recent research by Cooper et al., which also found elevated cortisol levels in men living with HIV in Africa [52].

## 5. Conclusions

The skin and mucosal layers play a critical role in the body’s immune defense, protecting against infections, inflammation, and hypersensitivity. However, this intricate system can be disrupted by infections such as HIV. HIV profoundly impacts the skin’s structure, immune functions, and microbiome, causing conditions like hyperplasia, inflammatory skin disorders, and hypersensitivity reactions. These effects are driven by CD4+ T cells depletion, cytokine imbalances, and disturbances in both innate and adaptive immunity. Notably, the skin microbiome in HIV patients undergoes significant shifts independent of CD4+ levels, indicating additional underlying mechanisms. Sex differences influence immune responses, with women exhibiting stronger defenses but higher autoimmune risks, while men experience a more rapid onset of immunosenescence with age. Environmental factors, including air pollution and heat, further worsen immune dysfunction and skin-related issues. Understanding and addressing these interconnected factors are essential for developing targeted treatments to enhance skin health and overall well-being in HIV patients.

## Figures and Tables

**Figure 1 ijms-26-03199-f001:**
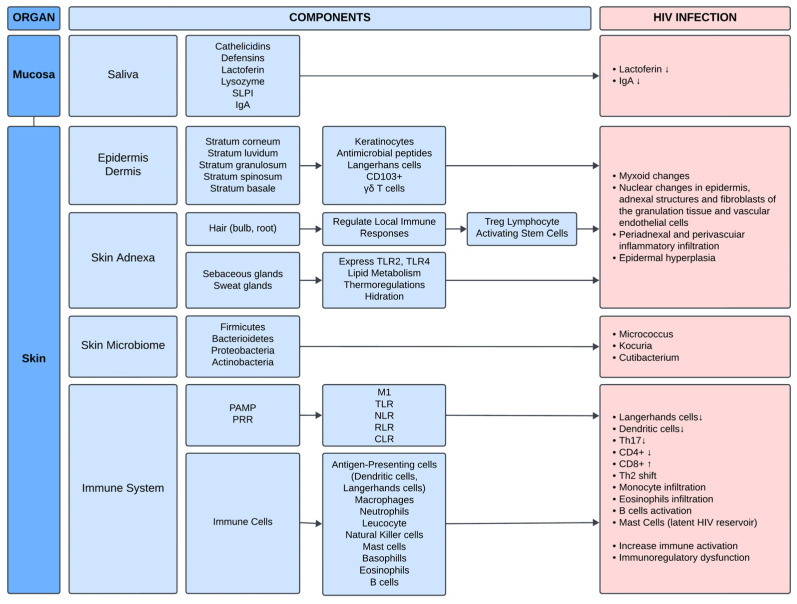
Skin barrier components in regards to the immune system and how HIV effects its function. ↑/↓: increased/decreased production; SLPI: Secretory leukocyte protease inhibitor; PAMP: pathogen-associated molecular patterns; PRR: pattern-recognition receptors; TLR: toll-like receptors; NLR: nucleotide-binding oligomerization domain-like receptors; RLR: retinoic acid-inducible gene I-like receptors; CLR: c-type lectin receptors.

## Data Availability

Not applicable.

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
