# Peer review of "HIV-Associated Dermatological Alterations: Barrier Dysfunction, Immune Impairment, and Microbiome Changes"

_ijms, 2025, doi:10.3390/ijms26073199_

Round 1
Reviewer 1 Report
Comments and Suggestions for Authors
The manuscript “HIV-Associated Dermatological Alterations: Barrier Dysfunction, Immune Impairment, and Microbiome Changes” reviews the different effects of HIV infection in several of the mucosal barriers, focusing on the skin as an immunological organ.
This review is well-written and easy for the reader to understand.
However some main points would improve this review. Please see bellow:
T cell exhaustion plays a great role in HIV pathogenesis and reservoir, I suggest adding a paragraph about how T cell exhaustion during HIV infection can affect the skin immune system at different time points.
This review is focused on the skin, but there are several paragraphs that just state general (already well known) HIV infection effects. Please relate all these to skin diseases. For example in “4.1” (Men and Women Differences), any of the information is related to the skin environment. I would strongly suggest refocusing them.
Same for “4.2” no mention of how the age can affect the dermatological diseases in HIV infection. Due to people living with HIV suffer more from skin frailty than healthy patients, how can this difference affect the skin environment?
The first paragraph of “4.4” focuses on respiratory epithelial barriers, this barrier impairment is not related at all to HIV infection. If the relation is due to low CD4+ count, if inflammation is due to CD8+ or excess of proinflammatory cytokines related to HIV infection, please state this instead of focusing on just general facts.
I think that also would be really helpful for the field to include studies of the different approaches that the field is working on to target specifically the mucosal barrier with IgA vaccines.
I suggest including more references to support the information provided in this revew.
Specific comments:
Lines 79-82: Add more studies and references to support the idea.
Line 95: Add a sentence stating why the reduction of these key component affect the mucosal environment.
Line 135-136: add reference
Line 142: add reference.
Figure 1: I suggest to add the function of the components in each compartment and relate to the outcome in HIV infection. In some of the cases is not clear yet but it would help the reader to relate how HIV infection could modify the compartments or how these compartments could be treated to reduce HIV infection burden.
Line 217-218: is this key theory supported by literature? If so add more references.
Line 228: It is clear that the CD4+ T cell depletion imbalance the immune response, but can you elaborate a little bit more on the direct impact on the skin. I suggest to add in this paragraph specific skin diseases related to low CD4 count, and specific opportunistic infections that can lead to further disease.
Lines 277-287: How these differences affect the skin environment in HIV infection. Is there any skin disease that is more prevalent in HIV infected females than males or viceversa?
Author Response
COMMENT REVIEWER 1
The manuscript “HIV-Associated Dermatological Alterations: Barrier Dysfunction, Immune Impairment, and Microbiome Changes” reviews the different effects of HIV infection in several of the mucosal barriers, focusing on the skin as an immunological organ.
This review is well-written and easy for the reader to understand.
However some main points would improve this review. Please see bellow:
General response: Thank you for your time and feedback. We agree that improving the literature and references would strengthen the manuscript. In response, we have added relevant references and extended the discussion in key areas. We hope these revisions address your concerns.
Comment 1: T cell exhaustion plays a great role in HIV pathogenesis and reservoir, I suggest adding a paragraph about how T cell exhaustion during HIV infection can affect the skin immune system at different time points.
Response 1: Thank you for your feedback. We have added a quick overview about T cell exhaustion in HIV pathogenesis in section 2.4.
Comment 2: This review is focused on the skin, but there are several paragraphs that just state general (already well known) HIV infection effects. Please relate all these to skin diseases. For example in “4.1” (Men and Women Differences), any of the information is related to the skin environment. I would strongly suggest refocusing them.
Response 2: Thank you for your valuable feedback. We recognize the limited availability of references specifically addressing HIV's immunological effects on the skin. To address this, we have revised section 4.1 and incorporated additional references to better emphasize the skin environment.
Comment 3: Same for “4.2” no mention of how the age can affect the dermatological diseases in HIV infection. Due to people living with HIV suffer more from skin frailty than healthy patients, how can this difference affect the skin environment?
Response 3: While finding age-specific dermatological immune effects in HIV was challenging, we have also added more references to section 4.2 to address skin frailty and its impact on the skin environment in people living with HIV.
Comment 4: The first paragraph of “4.4” focuses on respiratory epithelial barriers, this barrier impairment is not related at all to HIV infection. If the relation is due to low CD4+ count, if inflammation is due to CD8+ or excess of proinflammatory cytokines related to HIV infection, please state this instead of focusing on just general facts.
Response 4: Thank you for mentioning this. Upon reviewing that paragraph, I realize that the paragraph didn’t clearly link environmental effect to HIV-specific mechanisms. My initial focus was on environmental effects to the immune system, but I’ve revised the section to better highlight skin-related immune dysregulation due to environmental influences.
Comment 5: I think that also would be really helpful for the field to include studies of the different approaches that the field is working on to target specifically the mucosal barrier with IgA vaccines.
Response 5: Thank you for your suggestion. We have incorporated references to support the role of IgA as a potential HIV vaccine in Section 2.2.
Comment 6: I suggest including more references to support the information provided in this revew.
Response 6: We have incorporated additional references into the review to enhance the robustness and depth of the discussion.
Specific comments:
Comment 7: Lines 79-82: Add more studies and references to support the idea.
Response 7: We have enhanced this paragraph and included additional data regarding the changes that occur in the dermis and epidermis of HIV patients.
Comment 8: Line 95: Add a sentence stating why the reduction of these key component affect the mucosal environment.
Response 8: We have provided a detailed explanation of these components and their effects on the mucosal environment and disease pathogenesis in Section 2.2.
Comment 9: Line 135-136: add reference
Response 9: We have added the reference, as this statement is derived from the same source cited at the end of the paragraph.
Comment 10: Line 142: add reference.
Response 10: We have included supporting sentences and additional references to strengthen the argument.
Comment 11: Figure 1: I suggest to add the function of the components in each compartment and relate to the outcome in HIV infection. In some of the cases is not clear yet but it would help the reader to relate how HIV infection could modify the compartments or how these compartments could be treated to reduce HIV infection burden.
Response 11: Thank you for your suggestion. However, following our discussion, we have already addressed this topic in the related section. This figure is intended to guide researchers and clinicians by highlighting key areas to explore when studying the immunological aspects of HIV, particularly in the skin and mucosal regions.
Comment 12: Line 217-218: is this key theory supported by literature? If so add more references.
Response 12: We have included supporting sentences and additional references to strengthen the argument.
Comment 13: Line 228: It is clear that the CD4+ T cell depletion imbalance the immune response, but can you elaborate a little bit more on the direct impact on the skin. I suggest to add in this paragraph specific skin diseases related to low CD4 count, and specific opportunistic infections that can lead to further disease.
Response 13: Thank you for pointing this out. We have added clarifying sentences and supporting references to strengthen this statement.
Comment 14: Lines 277-287: How these differences affect the skin environment in HIV infection. Is there any skin disease that is more prevalent in HIV infected females than males or viceversa?
Response 14: Thank you for pointing this out. We have also added clarifying sentences and supporting references to strengthen this statement.
Reviewer 2 Report
Comments and Suggestions for Authors
The review manuscript addresses a problem that is still important today and affects many people, namely HIV-associated dermatological alterations. The structure is logical and the figures are informative. The authors have identified 32 literature references for the manuscript. In my opinion, the number could be increased. After reviewing the manuscript, I have the following questions that could be added to the manuscript:
Where do the authors see the Research Gaps on this topic?
What technical approaches are currently available to investigate the disease characterization? (e.g. single-cell RNA sequencing (scRNA-seq)
Do you have any information on how the conditions of HIV-infected skin vary with different skin types?
Author Response
COMMENT REVIEWER 2
The review manuscript addresses a problem that is still important today and affects many people, namely HIV-associated dermatological alterations. The structure is logical and the figures are informative. The authors have identified 32 literature references for the manuscript. In my opinion, the number could be increased. After reviewing the manuscript, I have the following questions that could be added to the manuscript:
General response: Dear reviewer, thank you for your time and feedback. We agree that expanding the literature base would strengthen the manuscript. In response, we have added relevant references and extended the discussion in key areas. We hope these revisions address your concerns.
Comment 1: Where do the authors see the Research Gaps on this topic?
Response 1: The research gaps we aim to address include understanding the role of immunological factors and microbiome analysis in dermatological diseases among HIV patients, as well as integrating these aspects to provide a more comprehensive understanding of disease mechanisms and progression. This review serves as a foundation for our main study exploring these correlations.
Comment 2: What technical approaches are currently available to investigate the disease characterization? (e.g. single-cell RNA sequencing (scRNA-seq)
Response 2: Thank you for your comment, we are utilizing 16S sequencing to analyze the skin microbiome and flow cytometry to characterize immune cells in HIV patients. These technical approaches allow us to investigate disease mechanisms by examining microbial composition and immune cell profiles, providing insights into the interplay between the microbiome and immunological factors in dermatological diseases.
Comment 3: Do you have any information on how the conditions of HIV-infected skin vary with different skin types?
Response 3: We are currently conducting a meta-analysis to explore the proportions of dermatological diseases in HIV patients across different factors such as age, sex, and geographical location. While this analysis does not specifically focus on skin types, it provides broader insights into how these conditions vary among diverse populations. This work is currently under review in another journal, and we anticipate that it will provide valuable insights for clinicians, enhancing their understanding of dermatological manifestations in HIV patients and informing better diagnostic and management strategies.
Round 2
Reviewer 1 Report
Comments and Suggestions for Authors
The manuscript “HIV-Associated Dermatological Alterations: Barrier Dysfunction, Immune Impairment, and Microbiome Changes” has improved after revision and it more clearly reviews the different effects of HIV infection in several of the mucosal barriers.
Author Response
Dear Reviewer,
Thank you for taking the time to review our manuscript. Your valuable suggestions have greatly contributed to its improvement, and we truly appreciate your insightful feedback.